# PROMPTING-BASED EFFICIENT TEMPORAL DOMAIN GENERALIZATION

## ABSTRACT

Machine learning traditionally assumes that training and testing data are distributed independently and identically. However, in many real-world settings, the data distribution can shift over time, leading to poor generalization of trained models in future time periods. Our paper presents a novel prompting-based approach to temporal domain generalization that is parameter-efficient, time-efficient, and does not require access to the target domain data (i.e., unseen future time periods) during training. Our method adapts a target pre-trained model to temporal drift by learning global prompts, domain-specific prompts, and drift-aware prompts that capture underlying temporal dynamics. It is compatible across diverse tasks, such as classification, regression, and time series forecasting, and sets a new state-of-the-art benchmark in temporal domain generalization. The code repository will be publicly shared.

## 1 INTRODUCTION

Machine learning has achieved great success in many applications in recent years, and most machine learning algorithms rely on the assumption that the training (i.e. source) and test (i.e. target) data are independently and identically distributed (i.i.d.). However, in reality, distribution shift and concept drift are often observed, and these non-i.i.d problems are more challenging to tackle. In domain adaptation (DA), extensive research has been conducted on adapting models to the target domain by modelling the domain relations between the source and the target Courty et al. (2016); Gong et al. (2012); Hoffman et al. (2014); Jimenez et al. (2019); Lao et al. (2020); Wang et al. (2020a); Yang & Hospedales (2016). However, such models assume that target domain data is available, which may not always hold in real-world settings. Domain generalization (DG) methods tackle the scenario where models are directly generalized to the target domain without the presence of the target data (labelled or unlabelled) Yue et al. (2019); Prakash et al. (2019); Shankar et al. (2018); Volpi et al. (2018); Hu et al. (2021b); Triantafillou et al. (2021); Kim et al. (2021); Wang et al. (2021a).

DG traditionally focuses on generalization among categorical-indexed domains with categorical task Wang et al. (2021c); Chen et al. (2022). In contrast, *temporal DG* addresses the continuously time-evolving distribution shift (namely concept drift) problem Bai et al. (2023); Nasery et al. (2021). For example, suppose we would like to predict house prices given information about the property's physical characteristics, such as square footage, number of bedrooms, number of bathrooms, and location. Since house prices are influenced by macroeconomic conditions and demographic trends that change over time, a regression model trained on data collected from the past few years could have poor predictive power next year Yin et al. (2022). However, suppose the macroeconomic and demographic factors change gradually over time. In that case, we can extrapolate their influence into the short-term future, and adapt the regression model to make more accurate predictions. Such cases are where temporal domain generalization can be applied. For example, suppose we know that the population in a particular country has been steadily aging over the past several years, which reduces the overall demand for many-bedroom houses. A temporal DG algorithm can anticipate that the demand will continue to fall for many-bedroom houses and adapt the price predictions for these houses accordingly: given the same features, a many-bedroom house next year will be priced some amount less than this year. Note that in the temporal DG setting, we do not get to see the "test domain", i.e., next year's house prices, during training. Therefore, temporal DG methods that model the continuously time-evolving data dynamics and generalize well to the future are needed.

Most standard DG methods cannot be directly applied to temporal DG. Different from standard DG problems, which aim to discover general representations among different domains and learn domain-invariant features, capturing the temporal dynamics of domain data changing over time is crucial for temporal DG. Learning domain-invariant features, namely time-invariant representations in temporal DG case, no longer work. Only a few methods studied temporal DG problem Nasery et al. (2021); Bai et al. (2023), which are inefficient and complex to be applied to large datasets and large models. Moreover, all the prior works only showed their effectiveness on classification and/or regression tasks, while missing demonstrations on other applications, such as time series forecasting. Therefore, a more efficient temporal DG framework enabling more diverse tasks is valuable.

Prompting is well-known for efficiently adapting a trained network to different tasks without retraining Lester et al. (2021); Vu et al. (2021); Gu et al. (2021); Li & Liang (2021); Asai et al. (2022); Wang et al. (2023). Most prior works Jia & Zhang (2022); Zhang et al. (2021); Li et al. (2022); Dunlap et al. (2022); Shu et al. (2023) adopting prompting for DG are applicable to only CLIP Radford et al. (2021) and cannot be applied to other architectures or tasks. PADA Ben-David et al. (2022) is a recent work proposed for DG. It first generates example-specific prompts, and then the generated prompts are applied to T5 for classification tasks. However, PADA is applicable only to classification tasks, and it can only generate word tokens as prompts.

Moreover, none of these prior works can generate time-sensitive prompts that capture temporal dynamics. In this paper, we proposed a parameter-efficient and time-efficient prompting-based temporal DG method. To capture temporal dynamics, domain-specific prompts are first generated on each domain. Then, our method learns time-sensitive prompts by modelling the temporal changes from domain-specific prompts and forecasts future prompts for unseen future domains. Our method also learns global prompts shared across all domains to learn generic representations. The prompts are generated in vector space and can be applied to a wide range of network architectures.

To sum up, our contributions are: (1) We propose the first prompting-based temporal DG method for addressing data distribution shifts over time. (2) Our method is parameter-efficient and time-efficient. In contrast to the state-of-the-art approach (Bai et al., 2023), which generates a full network for each domain, including the target domain, only a few parameters shared across all domains are allocated for prompt generation, and no additional parameters are needed for the target domain. (3) Our method is general and can be applied to many applications, including classification, regression, and time series forecasting.

## 2 RELATED WORK

**Domain generalization and adaptation** are research fields that have garnered significant attention in recent years due to their practical significance in real-world applications Ganin & Lempitsky (2015); Tzeng et al. (2017); Tremblay et al. (2018); Shankar et al. (2018); Volpi et al. (2018); Zhou et al. (2020). The primary goal of domain adaptation (DA) is to tailor models to specific target domains, using the similarities that exist between these domains Ben-David et al. (2010); Wang & Deng (2018). Continuous domain adaptation, a subset of DA, addresses the adaptation to domains characterized by continuous variables Hoffman et al. (2014); Jimenez et al. (2019); Lao et al. (2020); Wang et al. (2020a); Yang & Hospedales (2016). This may include temporal domain adaptation, which deals with domains that evolve over time. For instance, Courty et al. (2016); Gong et al. (2012) adapted their training loss to account for future data derived from prior domains. Similarly, the method proposed by Mancini et al. (2019) involves time-sensitive deep neural network parameters to control their evolution over time. Their network possesses domain-specific and domain-generic parameters, with the former integrating an added constraint that considers the similarity between domains. Meanwhile, other approaches like Wang et al. (2020a); Ganin et al. (2016) focus on learning time-invariant representations using adversarial methods.

Domain generalization (DG) methods build upon the insights from domain adaptation (DA) and aim to enhance the generalization capability of models across unseen (target) domains, where the data distribution may differ significantly from the source domain. These methods are crucial when adaptation approaches, like domain adaptation (DA), are not feasible due to unavailable target domain data or other possible limitations in adapting the base model. DG techniques encompass a range of strategies, as outlined in Wang et al. (2021c). DG methods can be categorized into three groups based on their focus. First, data manipulation methods, which include data augmentation

by manipulating input data through domain randomization Yue et al. (2019); Prakash et al. (2019), adversarial data augmentation Shankar et al. (2018); Volpi et al. (2018); Nazari & Kovashka (2020); Khirodkar et al. (2019) and data generation Qiao et al. (2020); Liu et al. (2018); Zhao et al. (2021); Garg et al. (2021). Second, representation learning by either applying domain-invariant representation learning techniques Deshmukh et al. (2019); Qi et al. (2021); Fan et al. (2021); Mitrovic et al. (2021) or feature disentanglement techniques Hu et al. (2021b); Triantafillou et al. (2021); Nam et al. (2021); Sun et al. (2021) to improve generalization. Third, learning strategy methods exploit various learning strategies like ensemble learning Wu & Gong (2021); Dubey et al. (2021), meta-learning Kim et al. (2021); Wang et al. (2021a), and gradient operations Tian et al. (2022); Rame et al. (2021) to enhance the overall generalization capability.

DG is essential for scenarios where domain adaptation comes short, and models must excel across unseen domains with diverse data distributions. However, most existing DG methods target categorical-indexed domains for categorical tasks. Temporal Domain Generalization (DG) is a lesser-explored area that deals with the ongoing changes in distribution, referred to as concept drift. Standard DG techniques aren't easily adjustable to handle temporal DG scenarios. Unlike regular DG, which aims for generalized representations across different domains, temporal DG focuses more on capturing the domain data's temporal dynamics. The GI Nasery et al. (2021) method uses adversarial training to generalize over time, altering the leaky ReLU activation for time dependence. However, its adversarial nature limits its efficiency with larger datasets or models. DRAIN Bai et al. (2023), a recent temporal DG approach, generates future model weights based on previous domains data but is inefficient in terms of parameters. Generating weights for state-of-the-art network architectures, like transformers, becomes challenging. Most existing works demonstrate efficacy only in classification and regression, neglecting other applications, underscoring the need for a more versatile temporal DG framework.

**Prompting Mechanism**: The concept of prompt-based learning has gained significant traction in the field of natural language processing (NLP) for adapting pre-trained language models (PLMs) to various downstream tasks. This framework involves conditioning the model with additional instructions to perform specific tasks. Elmo (Peters et al. (2018)), Bert (Devlin et al. (2018)), and Brown et al. (2020) introduced the approach of fine-tuning PLMs for downstream tasks through fixed prompting functions. This technique has succeeded particularly in few-shot classification tasks like sentiment analysis and natural language inference (Gao et al. (2021); Liu et al. (2021b), where manually designed prompts were employed.

However, formulating such a prompting function is challenging and often demands heuristic knowledge. In response to this challenge, recent efforts such as soft prompts (Lester et al. (2021); Vu et al. (2021); Gu et al. (2021)), P-tuning V2 (Liu et al. (2021a)), and prefix tuning (Li & Liang (2021)) have been made to treat prompts as adaptable parameters. It is worth noting that prompts encapsulate task-specific supervision with notably fewer supplementary parameters than competing techniques, such as Adapter (Wang et al. (2020b); Pfeiffer et al. (2020) and LoRA (Hu et al. (2021a)).

A different yet related angle to this topic is the casting of language modelling as a sequence-to-sequence task. This approach employs full transformer models, like the encoder-decoder paradigm, to autoregressively generate masked or altered token spans from input sequences (Raffel et al. (2020); Lewis et al. (2020)). The T5 model, introduced by Raffel et al. (2020), exemplifies this concept by treating every task as generative, where tasks are prefixed with a specific phrase to denote the operation. This approach has spiked different exploration across numerous areas, from adapting language models for diverse utilities (Brown et al. (2020)), extracting sentiment or theme-centric details (Jiang et al. (2020); Sun & Lai (2020); Shin et al. (2020); Haviv et al. (2021)), enhancing fine-tuning efficiencies ( Li & Liang (2021); Scao & Rush (2021), to functioning as few-shot learning techniques (Gao et al. (2021); Schick & Schütze (2021)).

Moreover, researchers have studied the transferability of prompts (Wang et al. (2021b); Vu et al. (2021); Su et al. (2021)), seeking to enhance the efficacy of prompt tuning across various tasks. Methods such as SPoT ( Vu et al. (2021)) choose a prompt based on a similarity metric, whereas ATTEMPT ( Asai et al. (2022)) incorporates an attention mechanism to draw from source prompts, initializing the prompt for its designated task. Wang et al. (2023) achieved a universal prompt by decomposing and distilling knowledge from source prompts. However, none of these approaches have considered the concept of temporal drift in their problem and have not been designed for DG where the target domain is unseen. This paper introduces a new prompting-based approach that is

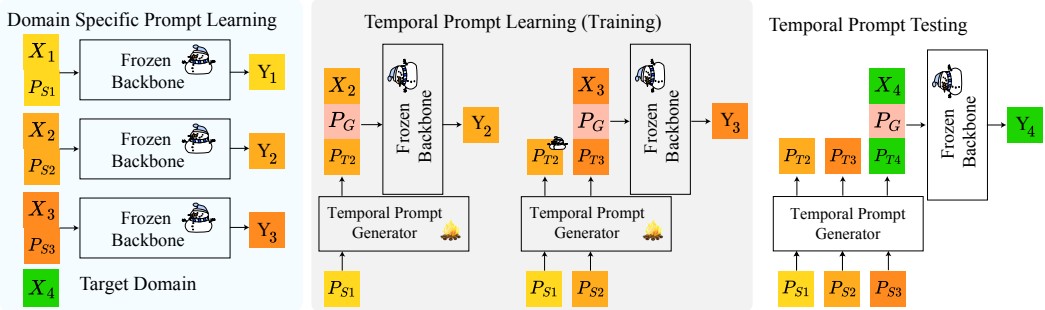

**Figure 1:** Overview of the proposed method. A set of source domains $D_1, D_2, D_3$ and a target domain $D_4$ are given. First, a backbone network is trained on the combined source domains in a pre-training phase. Then, domain-specific prompts $P_{S1}, P_{S2}, P_{S3}$ are learned independently on each source domain (while keeping the backbone network frozen) to learn the characteristics of each indexed domain separately. Next, a temporal prompt generator is trained to transform the domain-specific prompts to temporal prompts $(P_{T2}, P_{T3}, P_{T4})$ which can capture the temporal dynamics and concept drifts within the sequence of domains. finally, to capture the general knowledge across all domains, the general prompts $P_G$ are learned. For inference, the combination $[P_{T4}; P_G, X]$ is fed to the frozen backbone to perform the task on the target domain $D_4$.

both parameter-efficient and time-efficient, designed for temporal DG. It creates domain-specific prompts to capture temporal dynamics and models time-sensitive changes, anticipating prompts for future unseen domains.

## 3 METHOD

We address the problem of adapting a pre-trained model to future time periods under a realistic setting where data distributions evolve over time. Denote a set of temporal domains by $\mathcal{D} = \{D_t\}$, where $\{D_t | 1 \leq t \leq \tau\}$ represents the source domains, and $\{D_t | t > \tau\}$ represents the target domains. For example, each temporal domain may contain data points for one year. Data points from target domains are only observed during test time. Our goal is to learn the temporal dynamics within the sequence of source domains that can be directly generalized to future unseen target domains. Our solution utilizes two types of learnable prompts: domain-specific prompts ($P_{S(t)}$) and temporal prompts ($P_{T(t)}$). The domain-specific prompts estimate the distribution $\mathcal{P}(Y_t | X_t)$ for each domain $t$, where $Y_t$ are outputs and $X_t$ are inputs. The temporal prompts aim to capture the dynamics associated with temporal drift, and are generated using the domain-specific prompts. In Figure 1, the left and middle subfigures illustrate the training procedure, and the right one depicts the inference step.

### 3.1 BACKBONE NETWORK PRE-TRAINING

We start with a transformer-based network represented as $f_\theta$ as the model backbone. This network is pre-trained on the combined datasets from all source domains and the goal is to train the $f_\theta$ maximizing the likelihood $\mathcal{P}_\theta(Y_{1:\tau} | X_{1:\tau})$. After pre-training, $f_\theta$ weights are fixed in all later steps.

### 3.2 DOMAIN-SPECIFIC PROMPT LEARNING

The backbone network in Section 3.1 was pre-trained on the data aggregated across all source domains, without considering the differences in the individual domains. Intuitively, the pre-trained network captures "average" or "general" knowledge and can fail to learn details that reflect particular domains. Therefore, we adopt prompts to capture domain-specific information. For each domain $t$, we prepend the input $X$ with a prompt $P_{S(t)}$, which are learnable parameters. The combined result, represented as $[P_{S(t)}; X]$, is then processed by the frozen backbone network ($f_\theta$), which was pretrained across all source domains. To learn prompt $P_{S(t)}$, the model is trained to maximize the likelihood $\mathcal{P}_\theta(Y_t | [P_{S(t)}; X_t])$ while freezing the pre-trained model parameters $\theta$.

Learning on each domain independently, we derive domain-specific prompts $P_{S1}, P_{S2}, ..., P_{S(\tau)}$, effectively condensing domain knowledge into a concise set of parameters. Formally, for an input sequence $X$, the domain-specific prompt is represented as $P_S \in \mathbb{R}^n$.

### 3.3 TEMPORAL PROMPT LEARNING

To capture concept drift over time, we employ a temporal prompt generator to encode the temporal dynamics into temporal prompts. This module takes in domain-specific prompts from source domains and produces future temporal prompts. Here, we utilize a single-layer transformer encoder module, denoted by $g_\omega$, as our temporal prompt generator. In order to incorporate information from the preceding domains, we apply sequential training. Starting from domain $t = 2$, for each domain $t$ the temporal prompt generator $g_\omega$ receives domain-specific prompts, $P_{S1}, P_{S2}, \ldots, P_{S(t-1)}$, as input tokens. It then uses those prompts to generate the temporal prompts $P_{T2}, P_{T3}, \ldots, P_{T(t)}$. Specifically, as shown in Equation 1, it generates the temporal prompt $P_{T(t)}$ for domain $t$ from previous domain-specific prompts.

$$P_{T(t)} = g_\omega(P_{S1:(t-1)}), \quad t = 2, \ldots, \tau + 1 \tag{1}$$

Moreover, to help capture generic information across all domains, we learn a general prompt $P_G \in \mathbb{R}^n$. Finally, the input $X$ from domain $t$ is prepended by the generic prompt $P_G$ and the temporal prompt $P_{T(t)} \in \mathbb{R}^n$. The result, represented as $[P_{T(t)}; P_G; X]$, is fed into the frozen backbone network $f_\theta$ which has been pre-trained on all the combined source domains as described in Section 3.1. Both $P_G$ and the temporal prompt generator $g_\omega$ are trained to maximize the likelihood $\mathcal{P}_\theta(Y_t | [P_{T(t)}; P_G; X_t])$, while keeping the backbone network $f_\theta$ frozen.

Temporal prompts $P_{T2}, P_{T3}, \ldots, P_{T(\tau+1)}$ effectively capture temporal drift and help the pre-trained network to adapt to changes in the data distribution over time, and to anticipate future changes by capturing temporal trends.

---

**Algorithm 1** Training Procedure

**Require:** Source domains $\{D_t | 1 \leq t \leq \tau\}$, Target domains $\{D_t | t > \tau\}$, Pre-trained model to adapt $f_\theta$ parameterized by $\theta$, Temporal prompt generator $g_\omega$ parameterized by $\omega$, Labeled data from source domains $D_1, D_2, \ldots, D_\tau$
**Ensure:** Domain-specific prompts $P_{S1}, P_{S2}, \ldots, P_{S\tau}$, Temporal prompts $P_{T2}, P_{T3}, \ldots, P_{T\tau+1}$ Generic prompt $P_G$
1: **procedure** DOMAINSPECIFICPROMPTLEARNING
2:     **for** each domain $D_t$ in $\{D_t | 1 \leq t \leq \tau\}$ **do**
3:         Prepend $X$ with $P_{S(t)}$
4:         Process combined input $[P_{S(t)}; X]$ using frozen backbone $f_\theta$
5:         Train model to maximize likelihood $\mathcal{P}_\theta(Y | [P_{S(t)}; X])$ with $\theta$ fixed
6:     **end for**
7:     Return domain-specific prompts $P_{S1}, P_{S2}, \ldots, P_{S\tau}$
8: **end procedure**
9: **procedure** TEMPORALPROMPTLEARNING
10:     Initialize the temporal prompt generator $g_\omega$
11:     **for** each domain $D_t$ in $\{D_t | 2 \leq t \leq \tau + 1\}$ **do**
12:         Provide prompts $P_{S1}, P_{S2}, \ldots, P_{S(t-1)}$ to temporal prompt generator $g_\omega$
13:         Generate temporal prompt $P_{T(t)}$
14:         Prepend input $X$ from domain $t$ with $P_G$ and $P_{T(t)}$
15:         Process input $[P_{T(t)}; P_G; X]$ using frozen backbone $f_\theta$
16:         Train model to maximize likelihood $\mathcal{P}_\theta(Y | [P_{T(t)}; P_G; X])$ with $\theta$ fixed
17:     **end for**
18: **end procedure**

---

## 3.4 INFERENCE TIME

During the inference, the model utilizes the domain-specific prompts $P_{S1}, P_{S2}, \ldots, P_{S(\tau)}$ and generates temporal prompts $P_{T2}, P_{T3}, \ldots, P_{T(\tau+1)}$. To perform the target domain task, the frozen backbone receives the input $[P_{T(\tau+1)}; P_G; X_t]$ and predicts the output.

# 4 EXPERIMENTS

## 4.1 IMPLEMENTATION DETAILS

We utilize the Adam optimizer Kingma & Ba (2014) and consistently set the learning rate to $1e-4$ across all datasets. Our system is implemented in PyTorch and runs on a workstation powered by a 2.10GHz Intel Xeon(R) Gold 6230 CPU with 20 cores, paired with an NVIDIA RTX 5000 GPU. For each dataset, we tune the hyperparameters based on the suggestions from Bai et al. (2023). Additional experiment settings and results (e.g., network architectures and additional ablation results) are provided in the appendix.

## 4.2 COMPETING METHODS

We compare our model with several state-of-the-art methods including temporal domain generalization methods DRAIN Bai et al. (2023) and GI (Nasery et al., 2021), continuous domain adaption methods CDOT (Ortiz-Jimenez et al., 2019) and CIDA (Wang et al., 2020a), and prompting method ATTEMPT Asai et al. (2022) to validate the effectiveness of our temporal prompts. It's important to highlight that the original DRAIN employs two fully connected layers (DRAIN-2FC) in both encoding and decoding functions to transform the latent representations between LSTM units. To potentially boost DRAIN's performance, we also explored using three and four linear layers in both encoding and decoding functions. We call these models DRAIN-3FC and DRAIN-4FC, respectively. DRAIN-Best refers to the model achieving the highest performance using these configurations for the encoding/decoding functions.

We also compare against several baseline methods that do not consider temporal drift, including 1) The Vanilla-MLP, the MLP-based backbone network from DRAIN Bai et al. (2023), which is trained on the combined source domains. 2) Vanilla-Transformer, our method's transformer-based backbone network, which is trained on the combination of all source domains.

## 4.3 SYNTHETIC DATA

In order to comprehensively evaluate our proposed framework, we constructed 4 synthetic datasets. The first 2 datasets derive from the Mackey–Glass equations Mackey & Glass (1977), as shown in Equation 2. The rest 2 datasets are predicated on Cosine waves, defined in Equation 3. For introducing temporal shift to the data, we employed two strategies: alternating the data directly or adding a variable cosine wave with varying phases and frequencies across domains.

$$x(t+1) = x(t) + \beta \frac{x(t-\sigma)}{1+x^n(t-\sigma)} - \gamma x(t), \quad \begin{cases} \beta = 0.2 \\ \gamma = 0.1 \\ n = 15 \\ \sigma = 18 \\ t_{max} = 2600 \end{cases}, \quad x(t) = 0.1 \; if \, t < 18 \quad (2)$$

$$x(t) = \cos\left(a + \frac{\pi h}{\alpha} t\right) + \cos\left(b + \frac{\pi}{\beta} t\right), \quad \begin{cases} \alpha = 100 \\ \beta = 13 \\ a = 40 \\ b = 10 \\ h = 1 \end{cases}, \quad 0 < t < 2600 \quad (3)$$

**Data alternation**: For Mackey–Glass data, we induced temporal shift by changing $\sigma = 8 + i \times 2$ for each domain "$i$". For Cosine waves we induced temporal shift by changing $a = i$ and $h = i + 1$ for each domain "$i$". More visualizations of synthetic datasets are shown in the appendix.

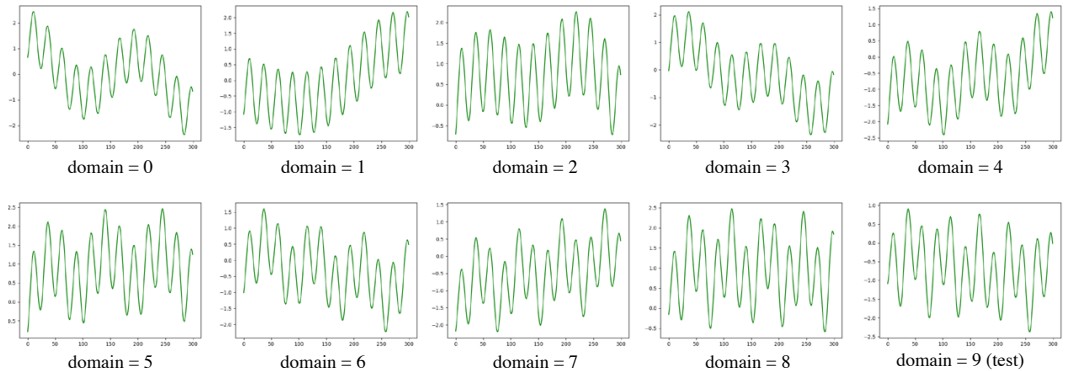

**Figure 2:** Applying temporal shift to Cosine waves time series by modifying Phase and Frequency, and adding an other variable cosine wave.

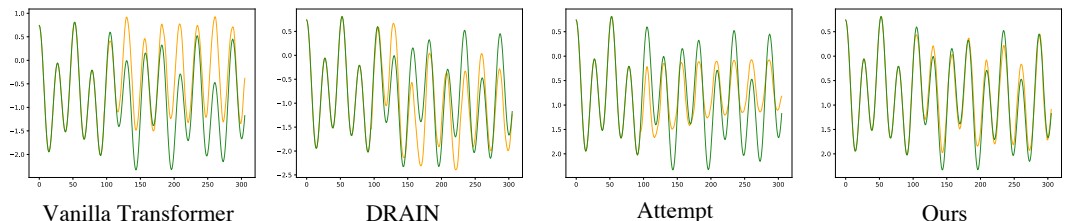

**Figure 3:** Qualitative results, on Sum of Cosine waves, when inducing temporal drift by data alternation and adding variable cosine wave.

**Adding variable Cosine wave**: For Mackey-glass time series, we add Eq 4 to our base equation 2 for each domain $i$ (see examples in figure 5). For the Cosine waves, we went one step further and add the same wave to the cosine wave after the data alternation (see examples in figure 6). More visualizations of the synthetic datasets are shown in the appendix.

$$0.5 \times \cos\left(100i + \frac{\pi(i+1)}{300}t\right) \tag{4}$$

Results on 4 synthetic datasets are summarized in Table 1, we also qualitatively visualize the results on the Cosine wave in Figure 3. Our proposed framework consistently outperformed the Vanilla Transformer, DRAIN, and Attempt models on synthetic data. Quantitatively, our model achieved the lowest MSE across both Mackey Glass and Sum of Cosine Waves datasets with either type of the temporal drifts. Qualitatively, it also demonstrates superior adaptability and accuracy.

**Table 1:** Comparison of our proposed framework, Vanilla transformer and other state of the arts using a synthetic data with 4 series generated based on Mackey Glass or Cosine waves.

| Base Data | Method | Data Alter. [MSE ↓] | Adding Cosine wave [MSE ↓] |
|---|---|---|---|
| Mackey Glass | DRAIN-Best | 0.1140 | 0.2164 |
| | Vanilla Transformer | 0.1315 | 0.2511 |
| | Attempt | 0.1278 | 0.2199 |
| | Ours | **0.0982** | **0.1975** |
| Sum of Cosine Waves | DRAIN-Best | 0.0085 | 0.2937 |
| | Vanilla Transformer | 0.0119 | 0.3708 |
| | Attempt | 0.0091 | 0.2974 |
| | Ours | **0.0068** | **0.2489** |

**Table 2:** Performance comparison of all methods in terms of classification error (in %) for classification tasks and mean absolute error (MAE) for regression tasks (both smaller the better.) Results of comparison methods on all datasets are reported from Bai et al. (2023). "-" denotes that the method could not converge on the specific dataset.

| Method | Classification [% error ↓] | | | Regression [MAE ↓] | |
|---|---|---|---|---|---|
| | 2-Moons | ONP | Elec2 | House | Appliance |
| Vanilla-MLP | $22.4 \pm 4.6$ | $33.8 \pm 0.6$ | $23.0 \pm 3.1$ | $11.0 \pm 0.36$ | $10.2 \pm 1.1$ |
| CDOT | $9.3 \pm 1.0$ | $34.1 \pm 0.0$ | $17.8 \pm 0.6$ | - | - |
| CIDA | $10.8 \pm 1.6$ | $34.7 \pm 0.6$ | $14.1 \pm 0.2$ | $9.7 \pm 0.06$ | $8.7 \pm 0.2$ |
| GI | $3.5 \pm 1.4$ | $36.4 \pm 0.8$ | $16.9 \pm 0.7$ | $9.6 \pm 0.02$ | $8.2 \pm 0.6$ |
| DRAIN Bai et al. (2023) | $\mathbf{3.2 \pm 1.2}$ | $38.3 \pm 1.2$ | $12.7 \pm 0.8$ | $9.3 \pm 0.14$ | $6.4 \pm 0.4$ |
| Vanilla-Transformer | $25.2 \pm 0.9$ | $33.6 \pm 0.5$ | $22.5 \pm 0.6$ | $11.8 \pm 0.3$ | $5.6 \pm 0.4$ |
| Attempt Asai et al. (2022) | $21.15 \pm 1.1$ | $34.10 \pm 0.6$ | $12.26 \pm 0.8$ | $9.0 \pm 0.4$ | $4.9 \pm 0.5$ |
| Ours | $8.1 \pm 1.0$ | $\mathbf{32.7 \pm 0.7}$ | $\mathbf{10.6 \pm 0.9}$ | $\mathbf{8.9 \pm 0.20}$ | $\mathbf{4.7 \pm 0.3}$ |

## 4.4 MAIN RESULTS

In this section, we evaluate the proposed model on a variety of datasets.

**Datasets**: A time series datasets: Crypto Arik et al. (2022); three classification datasets: Rotated Moons (2-Moons) Nasery et al. (2021), Online News Popularity (ONP) Ben-David et al. (2010), Electrical Demand (Elec2) Nasery et al. (2021); and two regression datasets: House prices (House) Nasery et al. (2021), Appliances energy prediction (Appliance) Bai et al. (2023).

For the classification and regression datasets (2-Moons, ONP, Elec2, House, and Appliance), we followed the procedure outlined in Bai et al. (2023) to partition the dataset into distinct temporal domains. The Crypto dataset contains 8 features on historical trades (e.g., open and close prices) for 14 crypto currencies. Our goal is to generate 15-step predictions for the 15-minute relative future returns (i.e., the target), with each step representing a 1-minute increment from the previous one. It starts from 2018 until 2021. We consider each month as one domain. We used the initial $90\%$ of entries from each month in 2018, 2019, and 2020 for training (across 36 domains), while reserving the remaining $10\%$ of entries for *in-domain* testing. The data from the first month of 2021 was designated for validation, with the subsequent three months of 2021 allocated for actual testing.

### 4.4.1 EXPERIMENTAL RESULTS

Table 2 summarizes the results in comparison to other state-of-the-art methods. The experiments are conducted 10 times for each method on every dataset, with both the mean and the standard deviation reported. It is observed that our proposed method yields lower errors in all instances except for the 2-Moons dataset. Notably, in the 2-Moons dataset, our method significantly outperforms the baselines but falls short when compared to the two recent domain generalization methods DRAIN Bai et al. (2023) and GI Nasery et al. (2021). This may be attributed to the low dimensionality of the 2-Moons dataset (only 2 dimensions), which leads to less generalizable backbones for prompt-based approaches (as evidenced by the poor performance in ATTEMPT as well).

Table 3 shows the time series forecasting results on Crypto dataset. To ensure a fair comparison, DRAIN, ATTEMPT, and our method all adopt the same backbone network Vanilla-Transformer. We explored two settings: one with fixed-length input sequences, and the other with variable-length input sequences. Our model is notably more accurate under both settings (with a lower RMSE) compared to DRAIN, Vanilla-Transformer, and ATTEMPT. Further, our method is significantly more parameter and time efficient than the current state-of-the-art temporal domain generalization method, DRAIN. While ATTEMPT, also a prompt-based approach, matches our efficiency in terms of parameters and time, it falls short in performance due to its inability to model temporal drift.

## 4.5 ABLATION STUDIES

First, we conduct ablation studies on the Crypto dataset and Elec2 datasets to see the impact of the proposed prompts. Table 4 shows that both two prompting mechanisms $P_T$ and $P_G$ contribute

**Table 3:** Performance comparison of our method against DRAIN Bai et al. (2023) and ATTEMPT Asai et al. (2022) on Crypto dataset in terms of root mean square error $\times 10^3$.

| len. | Method | #Parameter | Training time (s) | In domain | $D_{t1}$ | $D_{t2}$ | $D_{t3}$ |
|---|---|---|---|---|---|---|---|
| Fixed | DRAIN-2FC | 8M | 1634 | 3.96 | 4.27 | 7.03 | 7.24 |
| | DRAIN-3FC | 239M | 2520 | 3.82 | 3.90 | 6.75 | 6.89 |
| | DRAIN-4FC | 254M | 2827 | 3.60 | 3.61 | 6.69 | 6.69 |
| | Vanilla-Trans. | 69K | 239 | 4.00 | 4.42 | 7.19 | 7.43 |
| | Attempt | 93K | 684 | 3.57 | 4.03 | 7.22 | 7.45 |
| | Attempt-m | 93K | 684 | 3.54 | 3.79 | 6.96 | 7.35 |
| | Ours | 94K | 717 | **3.44** | **3.53** | **6.61** | **6.74** |
| Not-Fixed | DRAIN-2FC | 8M | 1634 | 4.97 | 5.22 | 7.78 | 7.98 |
| | DRAIN-3FC | 239M | 2520 | 4.61 | 4.95 | 7.38 | 7.47 |
| | DRAIN-4FC | 254M | 2827 | 3.66 | 3.74 | 6.82 | 7.03 |
| | Vanilla-Trans. | 69k | 239 | 4.08 | 4.44 | 7.28 | 7.55 |
| | Attempt | 93K | 684 | 3.85 | 4.29 | 7.51 | 7.75 |
| | Attempt-m | 93K | 684 | 3.79 | 4.12 | 7.16 | 7.43 |
| | Ours | 94K | 717 | **3.53** | **3.57** | **6.66** | **6.89** |

to better performance. Next, in order to study the impact of the number of training domains on our model performance, we conduct another ablation study on Mackey-Glass synthetic data(MG) with varying numbers of training domains as shown in Table 5. It is observed that our model's performance improves as the number of source domains increases, as a greater number of observed source domains make temporal patterns more evident.

**Table 4:** Ablation of effect of $P_G, P_T$ using Crypto and Elec2 dataset. ✓indicates the prompt being used.

| $P_G$ | $P_T$ | Crypto [RMSE $\times 10^3$ ↓] | | | Elec2 [MAE ↓] |
|---|---|---|---|---|---|
| | | $D_{t1}$ | $D_{t2}$ | $D_{t3}$ | $D_t$ |
| ✓ | | 3.57 | 6.66 | 6.84 | 14.9 |
| | ✓ | 3.53 | 6.71 | 6.80 | 14.7 |
| ✓ | ✓ | 3.53 | 6.61 | 6.74 | 10.6 |

**Table 5:** Impact of number of training domains on vanilla transformer and temporal prompting.

| Number of training domains | Data Alter (MG) [MSE ↓] | | Adding Cos. (MG) [MSE ↓] | |
|---|---|---|---|---|
| | Vanilla Trans. | Ours | Vanilla Trans. | Ours |
| 4 | 0.1818 | 0.1305 | 0.3007 | 0.2581 |
| 19 | 0.0877 | 0.0787 | 0.3326 | 0.2547 |
| 49 | 0.0930 | 0.0739 | 0.1645 | 0.1440 |

## 5 CONCLUSION

The efficacy of machine learning often depends on the assumptions that training and testing data are distributed independently and identically, an assumption that can be challenged by distribution shifts and concept drifts. This paper studied the scenarios where data distribution evolves over time. Such temporal drifts emphasize the need for temporal domain generalization (DG). In this paper, we propose a parameter and time-efficient prompting-based Temporal DG method that adeptly adapts pre-trained models to unforeseen future domains across various tasks, encompassing classification, regression, and time series forecasting. This represents a significant stride toward anticipating and adapting models to future domains using previous domains information.

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

## A APPENDIX

## B NETWORK ARCHITECTURES AND EXPERIMENTATION DETAILS

Below, we detail the architecture and other specific experiment details for each dataset.

**Architecture of frozen backbone network**: We choose backbones for each dataset to enable a fair comparison with state-of-the-art methods.

For the time series dataset **Crypto**, the initial inputs are passed through a linear layer, resulting in 64-dimensional embeddings. These embeddings are then processed by a transformer encoder layer. The transformer comprises a single encoder layer with four heads, and the hidden layers with dimensionality of 128. Finally, the output is passed through another linear layer to achieve the desired output size. We utilize the Mean Squared Error (MSE) loss for Crypto dataset.

For the datasets that are reported in DRAIN (Bai et al., 2023), the initial inputs for **Elec2**, **2Moons**, **House**, and **Appliance** are transformed through a linear layer to produce 128-dimensional embeddings, whereas for **ONP** it is a 32-dimensional embedding. These embeddings are subsequently processed by a transformer encoder layer. Notably, to align closely with the DRAIN paper's structure, our transformer encoder employs just one linear layer in the feed-forward segment, as opposed to the conventional two. The transformer setup involves a single encoder layer with one head. The hidden layers maintain a 128-dimensional structure for all datasets, with the exception of **ONP**, which is set at 64. Outputs are then channeled through another linear layer to derive the desired size. For regression datasets, we adopt the Mean Squared Error (MSE) loss, and for classification datasets, we use binary cross-entropy loss.

**Domain-specific prompts**: Domain-specific prompts are learnable parameters, whose sizes match the embedding dimensions for each dataset.

**Temporal prompt generator**: We employ a transformer with a single encoder layer and 1 heads as our temporal prompt generator. The transformer's hidden layers have a consistent 128-dimensional configuration.

### B.1 NON-SEQUENTIAL TEMPORAL PROMPT LEARNING

In the main paper, temporal prompts are generated sequentially. However, an alternative method exists to generate them non-sequentially. In this approach, we opt for a non-sequential training paradigm, wherein the model is exposed to all source domains simultaneously during the training process. To be precise, the temporal prompt generator, denoted as $g_\omega$, takes all domain-specific prompts $P_{S1}, P_{S2}, \ldots, P_{S(\tau)}$, and generates temporal prompts $P_{T2}, P_{T3}, \ldots, P_{T(\tau+1)}$. Table 6 compares performance of sequential temporal prompt generation vs non-sequential prompt generation, as it can be seen performance is on par with the main method and performance wise it is hard to say which method is superior.

**Table 6:** Comparing sequential temporal prompt generation vs non-sequential one.

| Method | Classification error [in % ↓ ] | | | Regression [MSE ↓] | |
| | 2-Moons | ONP | Elec2 | House | Appliance |
|---|---|---|---|---|---|
| Vanilla-Transformer | $25.2 \pm 0.9$ | $33.6 \pm 0.5$ | $22.5 \pm 0.6$ | $11.8 \pm 0.3$ | $5.6 \pm 0.4$ |
| Attempt Asai et al. (2022) | $21.15 \pm 1.1$ | $34.10 \pm 0.6$ | $12.26 \pm 0.8$ | $9.0 \pm 0.4$ | $4.9 \pm 0.5$ |
| Ours | $\mathbf{8.1 \pm 1.0}$ | $32.7 \pm 0.7$ | $\mathbf{10.6 \pm 0.9}$ | $8.9 \pm 0.20$ | $\mathbf{4.7 \pm 0.3}$ |
| Ours (not sequential) | $8.4 \pm 0.9$ | $\mathbf{31.8 \pm 0.7}$ | $11.2 \pm 0.8$ | $\mathbf{8.6 \pm 0.14}$ | $4.9 \pm 0.4$ |

### B.2 IMPACT OF EMBEDDING AND PROMPT SIZE ON MODEL PERFORMANCE

Table 7 shows ablations on embedding and prompt size. It is observed that for Crypto dataset, embedding/prompting size 64 and 128 provide similar better performance, and smaller embedding/prompting size results in a more parameter-efficient network, and 64 is selected for better model size and performance tradeoff.)

**Table 7:** Ablation of effect of prompt size and embedding size using Crypto dataset, in terms of root mean square error $\times 10$.

| Prompt size & Embedding size | Vanilla Transformer | | | Temporal prompting | | |
|:---:|:---:|:---:|:---:|:---:|:---:|:---:|
| | $D_{t1}$ | $D_{t2}$ | $D_{t3}$ | $D_{t1}$ | $D_{t2}$ | $D_{t3}$ |
| 32 | 4.20 | 7.20 | 7.45 | 3.57 | 6.64 | 6.85 |
| 64 | 4.42 | 7.19 | 7.43 | 3.53 | 6.61 | 6.74 |
| 128 | 4.52 | 7.59 | 7.79 | 3.45 | 6.58 | 6.79 |
| 256 | 4.45 | 7.25 | 7.39 | 3.45 | 6.64 | 6.79 |

### B.3 IMPACT OF TEMPORAL PROMOTING MODULE LAYERS ON MODEL PERFORMANCE

We used Mackey Glass data(MG) same as section 4.3 and studied the effect of number of layers in temporal prompt generation module in model performance, Table 8 present the results.

**Table 8:** Impact of temporal promoting module layers on model performance terms of MSE.

| Number of training domains | Data Alter (MG) | Adding Cosine wave (MG) |
|:---:|:---:|:---:|
| Vanilla Transformer | 0.1315 | 0.2511 |
| 1 | 0.0982 | 0.1975 |
| 2 | 0.0950 | 0.2053 |
| 3 | 0.1022 | 0.2119 |

### B.4 VISUALIZATION OF SYNTHETIC DATA

In this section we provided the visualization of synthetic data from section 4.3

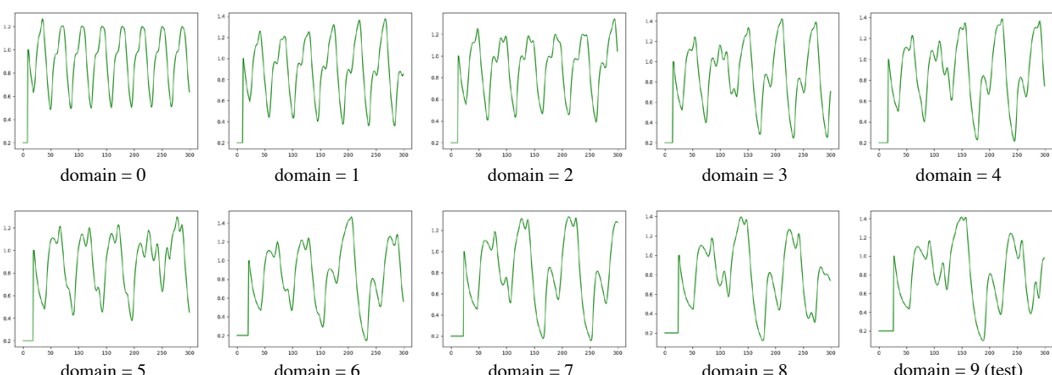

**Figure 4:** Applying temporal shift to Mackey-glass time series by modifying $\sigma$.

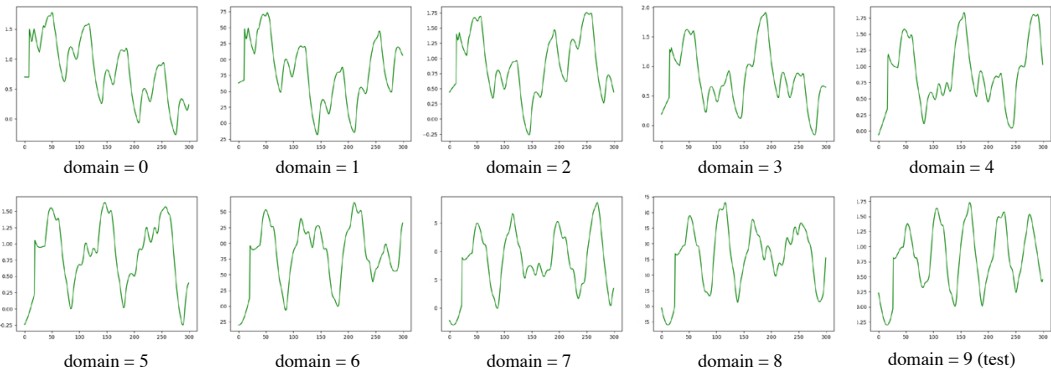

**Figure 5:** Applying temporal shift to Mackey-glass time series by adding Cosine wave.

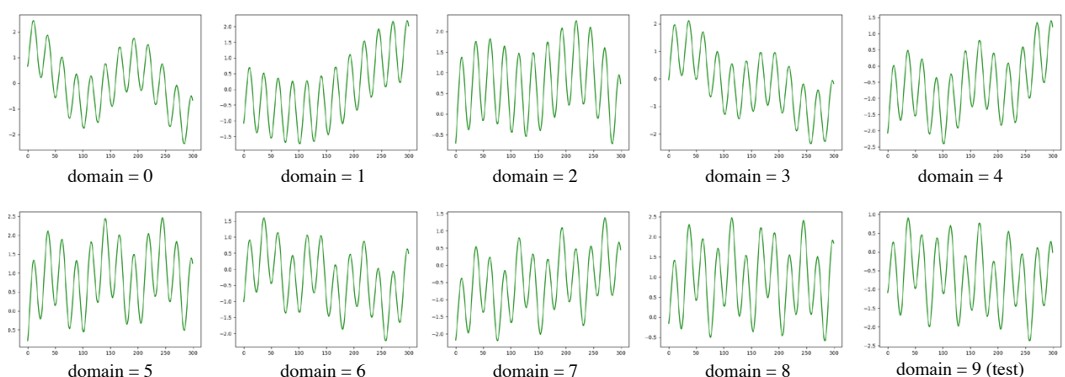

**Figure 6:** Applying temporal shift to Cosine waves time series by modifying Phase and Frequency, and adding an other variable cosine wave.

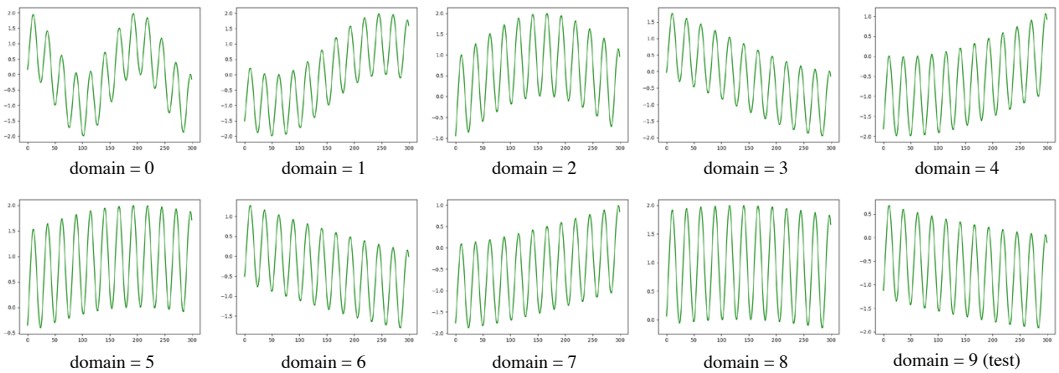

**Figure 7:** Applying temporal shift to Cosine waves time series by modifying Phase and Frequency.

