# OpenReview forum: "Prompting-based Efficient Temporal Domain Generalization"
_ICLR.cc/2024/Conference — Submitted to ICLR 2024_

### Official Review · Reviewer_ci1c · 2023-10-31

**Soundness:** 2 fair
**Presentation:** 3 good
**Contribution:** 2 fair
**Rating:** 5
**Confidence:** 2

**Summary:**

The paper describes an interesting prompt-learning strategy to capture temporal drifts in data over time. By representing data as occurring from different domains over time, the algorithm proposed by the authors learn two types of prompts from data by predicting future domains: One prompt capturing generalization across domains over time, and another prompt that incorporates ordered domain-specific prompts over time to capture temporal qualities exhibited by the data generation process. The authors empirically evaluate the proposed method over synthetic and real-world datasets to show the efficacy of the solution over other competing methods. Furthermore, they also provide an insightful ablation study to justify how prompts learned in the solution is useful in capturing temporal dynamics in data.

**Strengths:**

1. The paper is well written and easy to follow. The algorithm description and figure depictions really help in understanding the concepts and contributions presented in the paper.
2. Empirical evaluation shows good performance on both synthetic and real-world data, when compared to other existing mechanisms.
3. The ablation study clearly shows the need for two prompt types proposed in the paper.

**Weaknesses:**

The problem description seems to promise more than what the eventual solution delivers. Particularly, the paper is positioned by the authors as domain generalization and promises a solution in a space where learnings from different domains can be utilized to capture information useful for predicting unknown target domains. However, after reading the solution and dataset description, this falls short of expectation as the authors focus on concept drift within the same dataset. Data from the same data generation process is divided into multiple windows, where each window is called a domain. So, data drifts indicated by the authors are within data drift over time. In the literature, there has been multiple articles published on concept drift or data drifts in general over the past few decades. For example, please see Lu, Jie, et al. "Learning under concept drift: A review." IEEE transactions on knowledge and data engineering 31.12 (2018): 2346-2363. With this context, it is not clear why data windows within a dataset is termed as "domains", where it is truly not from a different domain. For true domain generalizability and adaptability, it would be good for the authors to explore how domain adaptation is setup, and empirically evaluate in-domain and across-domain generalizability and adaptation.

**Questions:**

A few elements of Algorithm 1 are not clear.
	1. Are the number of data points in each domain the same?
	2. In Step13, how exactly is PT(t) generated? Is is a concatenation of previous domain-specific prompts concatenated when provided as input to gw? Particularly, what is the difference between Line 12 and Line 13?
	3. Given my understanding of the problem setup, it is unclear what exactly is Y? Say in your housing price prediction example, is Y the house prices in the target domain (validation data) or house prices at domain t available in the training data?

The empirical evaluation in Table 2 and 3 shows that the proposed method has the least error across all datasets with greater than 2 variables. However, both the synthetic data generation and temporal drifts learned by the prompts seem to work for non-abrupt changes. Does this also work for abrupt drift?

---

> ### Author Response · Authors · 2023-11-21
>
> We thank the reviewer for your positive, insightful and valuable comments and suggestions which are very crucial for improving the quality of our manuscript.
>
> **The paper's problem description promises a domain generalization solution, but the actual solution seems to focus only on concept drift within the same dataset, falling short of initial expectations.**
> >This paper focuses on temporal domain generalization, addressing data distribution shifts over time. We followed the experimental setup of existing state-of-the-art temporal domain generalization methods [Nasery et al. (2021), Bai et al. (2023), Wang et al. (2020a), Ortiz-Jimenez et al.( 2019)]. And we cited the state-of-the-art method results in Table 1. We are not the first to propose temporal DG, and we followed the standard practice.
>
> ---
>
> **Clarification on the classification of different windows in a dataset as 'domains' when they represent data drift over time, suggesting a need for true domain generalizability and adaptability evaluation, as per existing literature on concept drift.**
>
> >We followed the experimental setup of existing state-of-the-art temporal domain generalization methods [Nasery et al. (2021), Bai et al. (2023), Wang et al. (2020a), Ortiz-Jimenez et al.( 2019)], and we split the dataset following the SOTA temporal DG methods. We are not the first to propose temporal DG, we didn't propose a new way to split the dataset. We followed the standard practice.
>
> ---
>
> **Clarification on elements of Algorithm 1, including uniformity in data points per domain, the generation process of temporal prompts, differences between Lines 12 and 13, and the specific nature of Y in the context of the housing price prediction example.**
>
> >The number of data points in each domain can vary. We do not enforce a uniform number of data points across all domains, as this depends on the dataset and the nature of the temporal segments being considered.
> >
> >In Line 12 of Algorithm 1, we are referring to the input tokens (PS1, PS2, ..., PS(t−1)) for the prompt generation module. These tokens represent the domain-specific prompts from previous time periods. In contrast, Line 13 describes the generation of PT(t), which is the current temporal prompt. This prompt is produced by the module using the concatenation of previous domain-specific prompts as input. The key distinction here is between the input prompts from previous domains (Line 12) and the output prompt for the current domain (Line 13).
> >
> >Y represents the label for the input data in the same domain. To illustrate with the housing price prediction example: if the features of a house in 2021 (X) correspond to a value of 200K, then Y (the label) would also be 200K. In other words, both the input (X) and output (Y) are derived from the same domain, ensuring that the model learns to predict within the context of each specific temporal domain.
>
> ---
>
> **Effectiveness of our Method in abrupt drift scenarios.**
> >Our model is optimized for gradual temporal drifts. If abrupt drifts are present in the training data, it's likely our model can adapt to similar abrupt changes during testing. However, without exposure to abrupt drifts in training, the model may not effectively handle sudden changes in the test data.

---

> > ### Author Response · Authors · 2023-11-22
> >
> > Dear Reviewer ci1c,
> >
> > We thank the reviewer for the time and effort you have dedicated to reviewing our paper. We aimed to address all your questions including (1) clarifying that the dataset this paper used (2) clarifying elements of Algorithm 1 and (3) the effectiveness of our method in abrupt drift scenarios.
> >
> > Could you please let us know if you have any further concerns? We are happy to address any further concerns you have. Any feedback would be highly appreciated. We look forward to hearing from you.

---

### Official Review · Reviewer_zvod · 2023-10-31

**Soundness:** 3 good
**Presentation:** 2 fair
**Contribution:** 2 fair
**Rating:** 3
**Confidence:** 4

**Summary:**

This paper introduces a new approach for adapting to these temporal changes without needing future data during training. Using a prompting-based method, it tweaks a pre-trained model to address time-related shifts by using different types of prompts that understand time-based patterns. This technique works for various tasks, like classification and forecasting, and achieves leading performance in adapting to time-based data changes.

**Strengths:**

1. The paper presents the first prompt-based method to handle temporal domain generalization.
2. The proposed method achieves better performance than existing methods in both accuracy and efficiency.
3. The studied problem is interesting and timely.

**Weaknesses:**

1. The motivation for using prompts still lacks proper motivation. Specifically, the motivation for using prompts is claimed as "none of these prior works can generate time-sensitive prompts that capture temporal dynamics." There are tons of other ways to learn temporal dynamics, and we don't have to use prompts.
2. The experiment setup also has some issues, e.g., the ablation study can be further improved, more baselines can be included, etc.
3. The overall presentation is a little messy. There are some undefined notations. The authors seem to misuse \citep{} and \citet{}, and the presented references impact the overall readability.

**Questions:**

1. What's the mathematical formulation of the prompt?
2. What would be the intuition of training the backbone network on the aggregated dataset? For some datasets with manually altered data distribution on each domain (like two moons), the decision boundary would be really difficult to learn if you mix all the data together.
3. Following the previous question, the ablation study can be further designed to remove the backbone model to verify if the backbone model is truly useful.
4. Some sentences are quite hard to understand, e.g., "For each domain $t$, we prepend the input $X$ with a prompt $PS(t)$, which are learnable parameters." Are both $X$ and $PS(t)$ learnable?
5. Not sure why some baselines are excluded in Table 3's comparison.
6. One of the major claims is also confusing: "Our paper presents a novel prompting-based approach to temporal domain generalization that does not require access to the target domain data". I feel like no access to the target domain data is a default rule of domain generalization.

---

> ### Author Response · Authors · 2023-11-21
>
> We thank the reviewer for your positive, insightful and valuable comments and suggestions which are very crucial for improving the quality of our manuscript.
>
> **What's the mathematical formulation of the prompt?**
>
> >Our work is inspired by the success of prompt-tuning in NLP tasks [Lester et al. (2021); Vu et al. (2021); Gu et al. (2021); Li & Liang (2021); Asai et al. (2022); Wang et al. (2023)]. As for the mathematical formulation of prompt, [Oymak et al. (2023)] developed a statistical foundation for gradient-driven prompt-tuning, examined its optimization and generalization behaviors, and investigated how it facilitates attending to context-relevant information. Additionally, they demonstrated that one-layer softmax-prompt-tuning is provably more effective than other methods, such as one-layer self-attention.
>
> >[Oymak et al. (2023)]: Oymak, S., Rawat, A.S., Soltanolkotabi, M. &amp; Thrampoulidis, C.. On the Role of Attention in Prompt-tuning. Proceedings of the 40th International Conference on Machine Learning (ICML 2023).
>
> ---
>
> **Doubts raised about the need for using prompts, considering the claim that other methods can't generate time-sensitive prompts for temporal dynamics, despite the existence of various alternatives to learn temporal dynamics.**
>
> >While alternative methods exist, our approach is well-supported and aligned with current research trends in the field. We have extended prompting from NLP to a lot more applications, which is truly valuable. Most previous prompting methods only demonstrated their efficacy in NLP tasks or NLP-related neural networks like GPT or CLIP. We proposed the first prompting-based temporal DG method, which is a generic model and is applied to various tasks. This shows great potential in using prompt learning for various applications other than NLP.
>
> ---
>
> **Typos and increasing readability.**
>
> >Thank you for your feedback. We will address the noted issues by correcting typos, clarifying undefined notations, and revising our citation format to improve readability and presentation.
>
>
> ---
>
> **The logic behind training the backbone network on an aggregated dataset, especially for datasets with manually altered distributions across domains, which could complicate learning the decision boundary.**
>
> >Through aggregating datasets, our network is designed to learn general/common features across the entire dataset, while domain-specific prompts target specific feature sets.
>
> ---
> **The ablation study can be further designed to remove the backbone model to verify if the backbone model is truly useful.**
>
> >Sorry, We didn't understand the question. How would prompting work without a backbone model? We followed the standard practice of prompting [Lester et al. (2021); Vu et al. (2021); Gu et al. (2021)], and a pre-trained backbone model is required.
>
> ---
> **Clarification on whether both and are learnable parameters in the context of pre-pending inputs with prompts for each domain.**
>
> >Prepending input with a prompt is a standard practice of prompting learning [Lester et al. (2021); Vu et al. (2021); Gu et al. (2021)]. In our model, the inputs are standard, non-learnable data from the dataset, similar to other classic models. Only the prompts are learnable parameters.
>
> ---
>
> **Missing baselines in Table 3.**
>
> >Other baselines are not applicable on Time series datasets. We will clarify that in Camera Ready.
>
> ---
> **Confusion about the claim stating that: "Our paper presents a novel prompting-based approach to temporal domain generalization that does not require access to the target domain data", as having no access to the target domain data is a default rule of domain generalization.**
> >Yes, we agree that no access to the target domain data is a default rule of domain generalization. We just want to emphasize it and highlight it, since it is the key difference between domain adaptation and domain generalization.

---

> > ### Comment · Reviewer_zvod · 2023-11-22
> >
> > Thanks for your feedback. Part of my concerns are addressed, but the major ones still remain.
> >
> > 1. Given the little improvement achieved compared to existing state-of-the-arts, I still wonder what is the rationale for using prompting in DG tasks. You use a bunch of papers of using prompting in language modeling/inference tasks. However, the prompting works for language models because the prompt is a piece of text inserted in the input examples, so that the original task can be formulated as a (masked) language modeling problem. Throughout the paper, it feels like you only claim you propose the first work of adopting prompting in temporal DG, and the model "is a generic model and is applied to various tasks", which I don't see why.
> >
> > 2. The design of your foundation model still does not make much sense to me. I still doubt training a model on the aggregated dataset can learn general features. Since the task is temporal DG, you can test whether there will be a performance drop of training your foundation model only on temporal domains that are close to the target domain. (ablation study)
> >
> > 3. Lastly, I am still not sure why important baselines are excluded in Table 3, and you seem to refuse to answer the question directly.

---

> > > ### Author Response · Authors · 2023-11-22
> > >
> > > We thank the reviewer for the comments and suggestions.
> > >
> > > **Given the little improvement achieved compared to existing state-of-the-arts**
> > > >We disagree that our improvement is little compared to existing SOTA methods. To improve efficiency, most machine learning methods trade performance for efficiency. While our methods show improved performance with much better efficiency.
> > >
> > > **I still wonder what is the rationale for using prompting in DG tasks.**
> > > >We focus on temporal DG, and there are only a few methods that studied the temporal DG problem and [Bai et al. (2023)] is the most recent and strong state-of-the-art method. These SOTA methods require training the whole or separate model for each temporal domain, which is inefficient in both parameter and time. Therefore, an efficient method is valuable. Prompting is well-known for efficiently adapting a trained network to different tasks in NLP tasks. Motivated by this observation, we explored methods adopting prompt-tuning for temporal DG problems.
> > >
> > > **It feels like you only claim you propose the first work of adopting prompting in temporal DG**
> > > >Most previous prompting methods only demonstrated their efficacy in NLP tasks or NLP-related neural networks like GPT or CLIP, and have rarely been applied to other architectures or tasks, such as classification, regression, and time series. Temporal DG is very different from NLP tasks, and it requires a different approach for modeling the backbone network and drift-aware prompts.
> > >
> > > **the model "is a generic model and is applied to various tasks", which I don't see why.**
> > > >Compared with both previous prompting methods and temporal DG methods, our method is generic. Compared with previous prompting methods, our method is more generic and shows great potential in using prompt learning for various applications other than NLP. Compared with previous temporal DG methods, we demonstrate the efficacy and efficiency of our model on more applications, including time series.
> > >
> > > **The design of your foundation model still does not make much sense to me. I still doubt training a model on the aggregated dataset can learn general features. Since the task is temporal DG, you can test whether there will be a performance drop of training your foundation model only on temporal domains that are close to the target domain. (ablation study)**
> > > >We don't understand compared with training on a single temporal domain, why training a model on more domains with more temporal patterns cannot learn more general features. The combined data itself contains more general features compared with a single domain. Given the limited time, we could only include the requested ablation study in camera-ready version, comparing our model with a baseline model training the backbone network on the last source domain. However, we want to emphasize that the performance of this baseline depends on whether the last domain is most similar to the target domain or not, while not depending on the generality of features. Training on combined source domain is the most robust and general way: it is robust to abrupt temporal changes in certain temporal domains and observes all temporal patterns across all temporal domains.
> > >
> > > **Lastly, I am still not sure why important baselines are excluded in Table 3, and you seem to refuse to answer the question directly.**
> > > >DRAIN [Bai et al. (2023)] is the most recent and strong state-of-the-art method, and we have provided a fair and thorough comparison with it in Table 3.
> > >
> > > > The baselines omitted are inherently unsuitable for time-series or sequential data analysis. The Crypto dataset, as shown in Table 3, is a time-series dataset. Applying methodologies not intended for sequential data to the Crypto dataset would not yield valid or meaningful comparisons.

---

### Official Review · Reviewer_HfUr · 2023-10-31

**Soundness:** 2 fair
**Presentation:** 3 good
**Contribution:** 2 fair
**Rating:** 5
**Confidence:** 4

**Summary:**

The authors propose a training paradigm for domain generalization. This contains 1) pre-training of a backbone model on all source domains, 2) the learning of source-domain-specific "prompts" for each source domain, 3) the learning of a "temporal prompt" for each source-domain, and 4) the learning of a global "prompt". At testing time, global and domain-specific prompts from the past can be used to make predictions in the new domain. The method is applied to time series classification and forecasting data sets and on a synthetic data set.

**Strengths:**

The paper is easy to follow and straightforward to read. The synthetic data experiment visually and numerically shows the shortcomings of other methods that motivate this work. Predictive performances are reported (at least partially) with standard deviations.

**Weaknesses:**

I am having difficulty thoroughly understanding how the architecture is trained in detail (more details in the question). I also question whether it is appropriate to split time series data artificially into different source domains. If a domain is not explicitly defined, one could pick an arbitrary period and define it as a domain, as was done on the crypto data set. There is no sound justification for why a one month period was chosen or why it would be better than a two week period. It furthermore seems that the presented method's performance largely lies in the confidence intervals of competing methods. I also miss some scientific curiosity about the learned prompt representations; there is much more potential in this work than reducing it to performance metrics.

**Questions:**

* Can you please clarify what the stopping criterion is when pre-training the backbone initially? You say the domain-specific prompts are learnable parameters, can you specify how they are connected to the backbone/output? You write they are concatenated, does this mean in the initial pre-training, we need to know the size of the prompt and mask the input accordingly? Or is there a linear layer whose parameters are learned? You might want to formalize all this by introducing a second set of parameters (as $\theta$ is always frozen). Furthermore, is a new temporal prompt generator trained for each temporal prompt, or is it reused? In Figure 1, you "freeze" $P_{T2}$ but not $P_{T3}$ in the next step, why? In Fig. 1's caption, you say, "[...] finally, [...] $P_G$ is trained". This implies a sequential training but from the figure, it seems like $P_G$ is also the output of the temporal prompt generator.
* Can you provide experimental results on data sets that naturally come from different domains?
    * If this is not the case, is it possible to use your method to **determine** the existence of different domains? I am thinking of comparing the learned domain-specific prompts, for example, in terms of their cosine similarity.
* Your algorithm omits many necessary details and does not add information that can't be inferred from the text. I would propose to either make the algorithm more informative (i.e., clarify the questions from above) or, to save space, remove it and clarify the points by extending Section 3.
* How do gradient-boosted trees perform on the regression tasks?
* How do standard forecasting methods such as ARIMA/Gaussian Processes perform on the data sets?
* Why are no standard deviations in Tables 3, 4, and 5 reported?
* Given its origin in NLP, I am not sure if "prompt" is the best fitting wording in the context of time series.
If all points can be addressed satisfactorily (particularly, the investigation into the representation of the learned prompts), I may consider raising my score.

---

> ### Author Response · Authors · 2023-11-21
>
> We thank the reviewer for your positive, insightful and valuable comments and suggestions which are very crucial for improving the quality of our manuscript.
>
> **Clarification on stopping criterion for pre-training the backbone.**
> >The pre-training process is halted when the validation loss fails to improve for a predefined number of epochs (3)
>
> ---
> **Clarification on how learnable domain-specific prompts are connected to the backbone/output, including details on concatenation, masking during pre-training, and potential use of a linear layer.**
> >We follow the standard practice of prompt learning to learn the prompts[Lester et al. (2021); Vu et al. (2021); Gu et al. (2021)], namely domain-specific prompts are extra tokens, and we are not increasing the dimension of tokens. The concatenation is about increasing the length of the input sequence, and transformers can take variable-length inputs.
> To answer your question, yes prompts are learnable parameters and they are part of the inputs inputting to the backbone network. No, we don't need to know the size of the prompt for pre-training.
>
> ---
> **Clarification on whether a new temporal prompt generator is trained for each prompt or if a single one is reused.**
> >For our temporal prompt generator, we adopt a dynamic tuning approach rather than training a new generator for each temporal prompt. Specifically, the process involves loading the weights from the previous step and then fine-tuning them based on the domain using previous domains information.
>
> ---
>
> **clarification on Figure one and and the sequential training versus simultaneous output from the temporal prompt generator.**
> >As explained in Section 3.3, it is sequential training. The temporal prompt generator is sequentially trained to generate prompts for the upcoming new domain, and previously learned prompts are fixed.
>
> ---
> **Concerns about the appropriateness of artificially splitting time series data into different source domains, with specific reference to the arbitrary period selection in the crypto dataset, lacking justification for the chosen duration.**
> >We followed state-of-the-art papers [Nasery et al. (2021), Bai et al. (2023), Wang et al. (2020a), Ortiz-Jimenez et al.( 2019) ] to split the dataset, and we split the Crypto dataset in the same way. We proposed one reasonable way of splitting the dataset, and we did not aim to propose the best way to split the dataset. Brute-force all kinds of ways splitting the dataset is a NP-hard problem, which is impossible and is also trivial for a paper.
>
> ---
> **Method's performance largely lies in the confidence intervals of competing methods.**
> >Yes, but our methods achieve the performances with a lot fewer parameters and much less training time. Normally machine learning models need to trade performance for efficiency, while our model shows great performance with better efficiency.
>
> ---
>
> **There is much more potential in this work than reducing it to performance metrics.**
>
> >We agree, so we expanded our works to more tasks and datasets. Most previous prompting methods only demonstrated their efficacy in NLP tasks or NLP-related neural networks like GPT or CLIP. We proposed the first prompting-based temporal DG method, which is a generic model and is applied to various tasks. This shows great potential in using prompt learning for various applications other than NLP.
> ---
> **Request for experimental results on datasets naturally from different domains, or if not available, whether your method can identify different domains, possibly by comparing learned domain-specific prompts through cosine similarity.**
>
> >For experimental results in Table 1, we followed state-of-the-art temporal DG methods [Nasery et al. (2021), Bai et al. (2023), Wang et al. (2020a), Ortiz-Jimenez et al.( 2019)] and adopted commonly used datasets. Crypto dataset is a time series dataset that naturally comes from different temporal domains. Domain-specific and/or temporal prompts may contain some information to determine the existence of different domains, while generic prompt cannot since it contains shared information across different domains. We will compare our domain-specific prompts in the camera-ready version.
>
> ---
>
> **Inquiry about the performance of gradient-boosted trees on regression tasks and standard forecasting methods like ARIMA/Gaussian Processes on the datasets.**
>
> >We have compared our methods with the current state-of-the-art methods. It would be great if the reviewer could kindly point us to existing state-of-the-art papers that apply these methods.
>
> ---
>
> **Standard deviations in Tables 3, 4, and 5.**
> >We will report that in Camera ready, they are similar to other tables.
>
> ---
>
> **Uncertainty regarding the suitability of the term "prompt" in the context of time series, considering its origin in NLP.**
> >We believe it is the best-fitting wording. We followed the exact standard practice of prompting [Lester et al. (2021); Vu et al. (2021); Gu et al. (2021)].
>
> ---

---

> > ### Comment · Reviewer_HfUr · 2023-11-22
> > **Thank you**
> >
> > I thank the authors for clarifying my concerns. I will raise my score by one.

---

### Official Review · Reviewer_Bw2Q · 2023-11-01

**Soundness:** 2 fair
**Presentation:** 3 good
**Contribution:** 2 fair
**Rating:** 5
**Confidence:** 4

**Summary:**

This paper introduces a new method for temporal domain generalization using prompts on transformer-based networks. This method is efficient and does not need data from future time periods during training. It uses global, domain-specific, and drift-aware prompts to adapt to data changes over time. The paper claims that the proposed method is adaptive on various tasks, such as classification, regression, and forecasting, The effectiveness of the framework is demonstrated through extensive experiments.

**Strengths:**

- This paper discusses a vital question on temporal domain generalization by leveraging soft prompts with transformer-based networks.
- The idea and motivation of this paper are easy to read. The idea and method proposed in this paper are clearly illustrated and introduced, making the reader easily understand.

**Weaknesses:**

- Overclaim 1. The second contribution of this work is "parameter-efficient and time-efficient". But their proposed method requires to train a transformer (Temporal Prompt Generator in Figure 1), which includes way more trainable parameters than existing methods such as DRAIN.
- Overclaim 2. As for the time-efficient aspect, there is no training time comparison analysis to demonstrate the claimed "time-efficiency." Especially, either pre-training or fine-tuning a transformer-based model to adapt to the specific task (temporal soft-prompt generation) are inefficient.
- Unfounded. The authors claim that "Only a few methods studied temporal DG problem Nasery et al. (2021); Bai et al. (2023), which are inefficient and complex to be applied to large datasets and large models," which is unfounded, no evidence supported, and without any quantitative analysis for demonstrating this assumption.
- The performance improvement is minor and not significant, especially since the proposed method achieves inferior performance than DRAIN (state-of-the-art of TDG) on the 2-moon dataset, a basic synthetic dataset on testing TDG. The performance of the proposed method is not convincing.
- The proposed framework seems to be adaptive on multiple modalities of transformer-based networks as the model backbone. However, the paper only evaluates their framework on one transformer-based network. The authors are highly encouraged to test their framework incorporated with multiple transformer-based networks.

**Questions:**

- ONP has been proven to obtain no domain shifting [1], which means most of the TDG-based methods are useless in ONP. However, the proposed methods, in contrast, obtain good performance on ONP. Is there any specific reason that can explain this phenomenon?

[1] Nasery et. al "Training for the Future: A Simple Gradient Interpolation Loss to Generalize Along Time
" NeurIPS 2021

---

> ### Author Response · Authors · 2023-11-21
>
> We thank the reviewer for your positive, insightful and valuable comments and suggestions which are very crucial for improving the quality of our manuscript.
>
>
> **Overclaim 1. How is the method "parameter-efficient and time-efficient" when it requires training a transformer with more parameters than existing methods like DRAIN?**
>
> >We appreciate the opportunity to address the concerns regarding any perceived overclaims in our paper. We would like to assure you that our assertions are carefully calibrated to reflect the findings of our research. As shown in Table 3, our approach, despite involving the training of a transformer (Temporal Prompt Generator), is actually more parameter-efficient. In fact, there are structural differences in how each method processes weights. DRAIN uses hypernetworks to generate weights for the backbone networks, and the size of hypernetworks is significantly higher than the backbone network. Especially when the domain task is a time series application, DRAIN requires generating weights for transformers, which requires an extremely high number of parameters. While for our method, whatever the domain task is, the goal is to generate a short prompting vector, which is much more parameter-efficient. Another aspect to note, for each domain DRAIN generates a backbone network, namely if there are $n$ domains DRAIN requires $n$ backbone networks. While for our method, the backbone network is shared, only a short vector (at most with length 128) is required for each domain. In general, time and parameter efficiency are the most well-known characteristics of prompt-tuning methods.
> ---
> **Overclaim 2. How is the method time-efficient when there's no comparative analysis of training time and both pre-training and fine-tuning a transformer for temporal soft-prompt generation are known to be inefficient?**
>
> >The assertion that our model lacks time-efficiency is addressed in Table 3, which presents a detailed comparison of training times between our method and existing approaches. This table clearly demonstrates that our model is not only efficient in terms of parameter usage, as previously discussed, but also in training time.
> ---
> **Unfounded. The authors claim that "Only a few methods studied temporal DG problem .... which are inefficient and complex to be applied to large datasets and large models," which is unfounded,  and without any quantitative analysis for demonstrating this assumption.**
>
> >The claim about the inefficiency and complexity of existing methods for temporal dynamic graph (DG) problems is backed by the data presented in Table 3. This table specifically illustrates how the number of parameters in the DRAIN method increases exponentially with time series datasets. Given that DRAIN relies on MLPs for encoding and decoding weights, this exponential increase in parameters makes it impractical for use with larger models.  Moreover, DRAIN requires generating weights for the backbone networks using hypernetworks, and generating weights for larger models is impractical.
> ---
> **The method shows only minor, insignificant performance improvements and underperforms compared to DRAIN on the basic 2-moon dataset, casting doubt on its effectiveness.**
>
> >We evaluated our method on 4 synthetic datasets and 6 real datasets. Our model demonstrates a significant improvement in performance compared to **DRAIN, which is the current SOTA model**, with an average margin of 20% across various benchmarks, as detailed in Tables 1, 2, and 3 on 6 real datasets and 3 synthetic dataset, and achieved comparable result on only 1 synthetic dataset. This general superiority is indicative of the model's robustness and effectiveness in various applications including classification, regression, and time series forecasting. Let alone our method is more time-efficient and parameter-efficient.
>
> ---
> **Applying proposed framework to various transformer-based networks.**
>
> >We appreciate the suggestion to evaluate our framework across multiple transformer-based networks. Indeed, our model is designed to be versatile and adaptable to a variety of models dealing with temporal data. While the current paper focuses on a single transformer-based network for a thorough and focused analysis, we recognize that using our model on other baselines can also be an interesting and Exploring this in future work is definitely something we find compelling and valuable.
>
> ---

---

> > ### Author Response · Authors · 2023-11-21
> >
> > **Given that ONP has no domain shifting, making most methods ineffective, what specific reasons explain the good performance of your methods on ONP?**
> > >Temporal DG methods widely applied on ONP. Contrary to the statement that ONP exhibits no domain shift, it  has minimal drift [1]. The notable performance of our model on ONP can be attributed to the integration of general prompts (PG) in our framework. These general prompts capture shared patterns across domains and enhance the model's ability to adapt and perform effectively even in stable environments with minimal drift. As demonstrated in Table 4, combining Temporal and General Prompts plays a crucial role in boosting performance. This dual-prompt approach allows our model to maintain high accuracy and adaptability in various scenarios, including ONP, where traditional TDG-based methods might not be as effective.
> >
> > ___

---

> > > ### Comment · Reviewer_Bw2Q · 2023-11-22
> > > **Thank you for the response**
> > >
> > > I appreciate the discussion and feedback from the authors, which addressed some of my questions and concerns. I will increase my score accordingly.

---

### Author Response · Authors · 2023-11-21

We thank all the reviewers for their valuable and insightful comments. We have addressed the reviewers points in our individual responses to each reviewer, and please let us know if there are any new questions.

---

### Author Response · Authors · 2023-11-22

We thank all reviewers for the valuable and positive feedback, and we truly appreciate the time and effort you have dedicated to reviewing our paper. We would like to take this opportunity to reemphasize the key contributions of our work and kindly request your reconsideration of the paper's overall score in light of these contributions.

Contributions:
To our knowledge, we propose the first prompting-based temporal domain generalization (DG) method for addressing data distribution shifts over time. This approach is not only versatile across various tasks but also demonstrates superior performance with significantly fewer parameters and reduced computational time when compared to existing methods as outlined in SOTA publications [Nasery et al. (2021), Bai et al. (2023), Wang et al. (2020a), Ortiz-Jimenez et al. (2019)].

Contrary to the common trade-off between efficiency and performance, our method exemplifies how prompt learning can enhance both parameter and time efficiency while simultaneously improving overall model performance. Further, while existing prompting methods have primarily demonstrated success in NLP tasks and associated neural networks, our research extends the applicability of prompt learning to a broader range of applications beyond NLP.

In light of these contributions, we kindly request your reconsideration of the overall score assigned to our paper. We understand the rigorous evaluation process. However, we believe that the outlined contributions warrant a thorough reevaluation of the paper's merit. We are more than willing to address any additional concerns or queries you may have and provide further clarification on specific points raised during the review process.

---

### Meta-Review · Area_Chair_e5AR · 2023-12-12

**Metareview:**

This easy to follow paper has been assessed by four knowledgeable reviewers. All of them recommended a rejection (including one straight and three marginal reject scores). The issues brought up by the reviewers reflected their confusion about the primary claims of the paper, and a number of detailed conceptual items. The authors have engaged the reviewers in extensive discussions which prompted two of the revivers to increase their ratings, but still no one recommended even marginal acceptance. This paper presents a promising work that could benefit from broader ablation studies and clearer presentation of the results and claims, but in its current shape it is not ready for inclusion in the ICLR 2024.

**Justification For Why Not Higher Score:**

No one reviewer voted for even marginal acceptance.

**Justification For Why Not Lower Score:**

n/a

---

### Decision · Program_Chairs · 2024-01-16

Reject